# Deep Learning-Based Automatic Detection of ASPECTS in Acute Ischemic Stroke: Improving Stroke Assessment on CT Scans

**DOI:** 10.3390/jcm11175159

**Published:** 2022-08-31

**Authors:** Pi-Ling Chiang, Shih-Yen Lin, Meng-Hsiang Chen, Yueh-Sheng Chen, Cheng-Kang Wang, Min-Chen Wu, Yii-Ting Huang, Meng-Yang Lee, Yong-Sheng Chen, Wei-Che Lin

**Affiliations:** 1Department of Diagnostic Radiology, Kaohsiung Chang Gung Memorial Hospital and Chang Gung University College of Medicine, Kaohsiung 83301, Taiwan; 2Department of Computer Science, National Yang Ming Chiao Tung University, Hsinchu 30010, Taiwan; 3Neurology Department, Pingtung Christian Hospital, Pingtung 90053, Taiwan; 4Department of Emergency Medicine, Kaohsiung Chang Gung Memorial Hospital and Chang Gung University College of Medicine, Kaohsiung 83301, Taiwan; 5Institute of Biomedical Engineering, National Yang Ming Chiao Tung University, Hsinchu 30010, Taiwan

**Keywords:** The Alberta stroke program early CT score, artificial intelligence, convolutional neural network, acute ischemic stroke

## Abstract

(1) Background: The Alberta Stroke Program Early CT Score (ASPECTS) is a standardized scoring tool used to evaluate the severity of acute ischemic stroke (AIS) on non-contrast CT (NCCT). Our aim in this study was to automate ASPECTS. (2) Methods: We utilized a total of 258 patient images with suspected AIS symptoms. Expert ASPECTS readings on NCCT were used as ground truths. A deep learning-based automatic detection (DLAD) algorithm was developed for automated ASPECTS scoring based on 168 training patient images using a convolutional neural network (CNN) architecture. An additional 90 testing patient images were used to evaluate the performance of the DLAD algorithm, which was then compared with ASPECTS readings on NCCT as performed by physicians. (3) Results: The sensitivity, specificity, and accuracy of DLAD for the prediction of ASPECTS were 65%, 82%, and 80%, respectively. These results demonstrate that the DLAD algorithm was not inferior to radiologist-read ASPECTS on NCCT. With the assistance of DLAD, the individual sensitivity of the ER physician, neurologist, and radiologist improved. (4) Conclusion: The proposed DLAD algorithm exhibits a reasonable ability for ASPECTS scoring on NCCT images in patients presenting with AIS symptoms. The DLAD algorithm could be a valuable tool to improve and accelerate the decision-making process of front-line physicians.

## 1. Introduction

Acute ischemic stroke (AIS) is major public health issue affecting populations globally. It is the most common cause of adult disabilities, resulting in a tremendous burden on national healthcare systems, including the National Health Insurance system in Taiwan. According to ASA guidelines [1], AIS patients with large vessel occlusion (LVO), who meet the clinical and image criteria, should receive intra-arterial thrombectomy (IAT) as soon as possible. The Alberta Stroke Program Early CT Score (ASPECTS) is a tool for the standardized evaluation of AIS severity of the anterior circulation [2,3], wherein a score of ≥6 is an important cutoff value and is one of the criteria qualifying patients for IAT. Nevertheless, the early image changes of a stroke event, including loss of the normal gray/white matter interface and effacement of the cortical sulci, may be subtle and particularly hard for clinicians to detect. It is, therefore, a challenge for clinicians to provide a prompt and accurate diagnosis of AIS, and to determine the optimal management protocol in an emergency scenario.

Over the past decade, artificial intelligence (AI) techniques have greatly impacted the field of stroke image analysis in terms of automating the diagnostic process, improving diagnostic accuracy, and enhancing predictions of prognosis [4]. In particular, automatic lesion identification or segmentation is one of the most important elements in precision medicine; as such, recently developed machine learning techniques have demonstrated promising results in the realm of automatic diagnosis. Notable advances include lesion segmentation of AIS with diffusion-weighted imaging (DWI) [5], development of the automatic ASPECT scoring system for the AIS area [5,6], detection of the hyperdense middle cerebral artery sign on CT [7], the automatic quantification of cerebral edema [8], and the prediction of the final shape of lesions [9]. In light of these advances, the automatic diagnosis and ASPECT scoring of AIS would be a valuable tool in an era where faster thrombolysis is recommended. However, there is currently no proposed AI model for automatic ASPECT scoring based on non-contrast CT (NCCT).

Time is of the essence when clinicians are faced with an AIS event. Meanwhile, it is essential for clinicians to take note of several qualifying criteria of IAT, including baseline performance of patients, clinical, and imaging assessments of disease severity, as well as the key factor, the time period between arterial occlusion and revascularization. Although we cannot influence the time it takes a patient to arrive at the hospital, we may be able to reduce the time required for imaging evaluation. Automatic lesion identification may expedite accurate diagnoses by non-specialists, and effectively accelerate the time to revascularization. In this study, we developed a deep learning-based automatic detection (DLAD) algorithm for AIS automated ASPECT scoring and compared it with the ASPECTS readings on NCCT as performed by radiologists and different clinicians.

## 2. Methods

### 2.1. Dataset and Image Annotation

We utilized a total of 258 patient images with suspected AIS symptoms presenting within 8 h of last known well (LKW) and initial brain NCCT (slice thickness = 5 mm) from a single medical center in Taiwan as the training and testing datasets. The 168 patient images included in the training dataset were collected from January 2016 to December 2018. The 90 patients included in the testing dataset were collected from August 2019 to April 2020. The protocol was approved by the research ethics committees of the hospital.

All NCCT brain images were reviewed, and the ground truths of ASPECTS were scored by two experienced neuroradiologists slice by slice, without blind to follow-up DWI images within 1 week. All annotated lesions were considered as true lesions based on the consensus of two neuroradiologists. For the NCCT template with ASPECTS regions, the locations and segmentation of ASPECTS regions in the MNI standard space were annotated manually by a single neuroradiologist using the ICBM152 brain template.

### 2.2. Development of the DLAD Algorithm for AIS

Each NCCT image underwent soft intensity cropping to mitigate the confounding effect of non-brain tissues such as bone and air, and the bounding volumes from individual ASPECTS regions were extracted from the intensity-cropped NCCT images. This is performed by manually delineating the region of interest (ROI) of ASPECTS regions in the ICBM152 brain template and inversely registering the ASPECTS ROIs to the NCCT images using the Segment and Deformation function in SPM12 (Ashburner et al., 2014). The deep learning model was then used to assess the presence of stroke in each bounding volume.

The deep learning model used in this study was based on a 2.5-dimensional convolutional neural network (CNN) architecture, which consists of a 2D slice encoder, a slice feature aggregator followed by a fully connected classifier. The slice encoder is a 7-layer 2D CNN whose purpose is to independently extract high-level image features from each slice. The feature maps of all slices were then concatenated and fed into to the feature aggregator, which is a 1-layer 3D CNN that summarizes the features across multiple slices. This yields the summarized features of the entire image volume. These voluminal features were then fed into a 2-layer fully connected classifier to generate the classification result (for detailed network configurations, see Table 1). The deep learning model was trained in a two-step manner. The first step was a pre-training step, where the slice encoder and the slice classifier were trained using the training data with slice-level annotations. In addition, to mitigate the potential inaccuracy caused by equivocal regions, all normal slices ipsilateral to any lesion were excluded from the training data when training the slice encoder. The second step was the fine-tuning step, where the slice encoder and slice classifier were initialized using the parameters obtained from pre-training, and the entire model was then trained using the data with regional-level annotations. The training process was performed using an Adam optimizer [10] for 200 epochs. Data augmentation with random rotation (±15 degrees for X and Y dimensions), translation (±2% image width for X and Y dimensions), and scaling (±10% for X and Y dimensions) were used during the training process (note that data augmentation excluded any spatial transformation along Z dimensions due to consideration of the poor interslice resolution). The batch size was set to 50, and the initial learning rate was set to 0.0005 with an exponential decay rate of 0.99 per epoch. The accuracy and loss value over the training data after each epoch are shown in Figure 1.

### 2.3. Assessment of DLAD and Physicians’ Performance

All 90 testing images were input into the DLAD algorithm and output with ASPECTS. To assess the performance of physicians with different subspecialties and levels of experience in reading ASPECTS, we enrolled one ER physician, one neurologist, one radiologist, and one neuroradiologist for the NCCT reading task. They received the first reading task on the testing dataset alone. To validate the clinical impact of the DLAD, these physicians received the second reading task on the same testing dataset assisted with ASPECTS of DLAD. The interval between the two reading tasks was at least 2 months. The expert reading of ASPECTS based on the NCCT were used as the ground truth.

The performance assessments of the DLAD and the physicians included: (1) sensitivity, specificity, accuracy, and F1 score; (2) agreement on each ASPECTS region and total ASPECT score were analyzed using the Cohen’s kappa coefficient and intraclass correlation coefficient (ICC); (3) the area under the curve (AUC) for individual region-level ASPECTS, total ASPECTS score, and the dichotomized ASPECTS (≥6 and <6); and (4) the permutation test of physician’s performance with DLAD and without DLAD (10.000 permutations, one-tailed). All statistical analyses were performed using SPSS statistical software (Version 23.0; IBM, Armonk, NY, USA) and MATLAB software (MathWorks, Inc., Natick, MA, USA). The threshold for statistically significant differences was defined as *p* < 0.05.

## 3. Results

### 3.1. Dataset Characteristic

For the 168 patient images included in the training dataset, the initial NCCT was performed within a mean time of 259 ± 166 min from LKW. For the 90 patients included in the testing dataset, the initial NCCT was performed within a mean time of 286 ± 209 min from LKW. These characteristics of both datasets were listed in Table 2.

### 3.2. Performance of DLAD

After applying the DLAD to the testing dataset of 90 patients, the sensitivity, specificity, accuracy, F1 score, kappa, and AUC of the prediction of infarction on all ASPECTS regions were 65.2%, 81.6%, 79.7%, 0.43, 0.32, and 0.73, respectively. Table 3 shows the performance of the DLAD on each ASPECTS region. Figure 2 shows the distribution of both human and the DLAD of ASPECTS scoring at each individual ground truth ASPECTS on NCCT.

### 3.3. Performance of Physicians without and with DLAD

Table 4 shows the individual performance of DLAD and each subspecialist on the testing dataset, which was compared with their performance with DLAD, in both ASPECTS regions and dichotomized ASPECTS tests.

In all ASPECTS regions, the mean sensitivity, specificity, accuracy, and F1 score of the physicians in the doctor-alone testing were 33.8%, 95.2%, 88.1%, and 0.38, respectively. The mean kappa value between the physicians and the ground truth was 0.32. After the DLAD was applied, the mean sensitivity, specificity, accuracy, and F1 score of the physicians in the DLAD-assisted test were 53.7%, 92.1%, 87.7%, and 0.50, respectively. The mean kappa value between the physicians with DLAD and the ground truth was 0.43.

In dichotomized ASPECTS (≥6 and <6), the mean sensitivity, specificity, accuracy, and F1 score of the physicians in the doctor-alone testing were 26.4%, 98.3%, 91.1%, and 0.34, respectively. The mean ICC between the subspecialists and the ground truth was 0.46. After the DLAD was applied, the mean sensitivity, specificity, accuracy, and F1 score of the physicians in the DLAD-assisted test were 52.8%, 94.9%, 90.7%, and 0.52, respectively. The mean ICC between the physicians with DLAD and the ground truth was 0.64.

The receiver operating characteristic (ROC) curves and the areas under the ROC curves (AUCs) were used to evaluate the performance of the DLAD model (Table 4 and Figure 3). The AUC of the ER physician, the neurologist, the radiologist, and the neuroradiologist in all ASPECTS regions were, respectively, 0.56, 0.63, 0.65, and 0.73 unassisted; and 0.69, 0.77, 0.72, and 0.73 with DLAD assistance, respectively. Of note, the specificity of each different unassisted subspecialist was better than DLAD. Importantly, the physicians’ performances improved with DLAD assistance, without sacrificing specificity. With 10,000 permutations, the respective performances of the ER physician, neurologist, and radiologist significantly improved with DLAD in both ASPECTS regions and dichotomized ASPECTS levels, especially regarding the sensitivity of AIS detection. The AUCs were significantly increased with the shifting of the ROC curve toward the top. The difference between neuroradiologist without and with DLAD was not significant.

## 4. Discussion

To summarize, the sensitivity, specificity, and accuracy of DLAD for prediction of ASPECTS were 65%, 82%, and 80%, respectively. These results reveal that the DLAD algorithm was not inferior to the radiologist-read ASPECTS on NCCT. Moreover, with the assistance of DLAD, the individual sensitivity of the ER physician, neurologist, and radiologist improved, while the ROC points of these subspecialists shifted toward the upper left corner.

Owing to moderate agreement between the readers in all ASPECTS regions [11,12], the ASPECTS evaluation is frequently dichotomized in literature. We used the dichotomized ASPECTS of ≥6 as it is an important cutoff value and one of the criteria for IAT. We noted significant differentiation between the subspecialists in terms of ASPECTS readings. Quite unsurprisingly, the impact of the DLAD algorithm was more significant for the subspecialists with less experience in reading ASPECTS; in addition, it did not significantly improve detection for an experienced neuroradiologist. The ground truths of ASPECTS were annotated on NCCT without blind to follow-up DWI images. The detection rates of stroke lesion may be more sensitive based on both NCCT and follow-up DWI images, which may account for why the neuroradiologist had a lower stroke detection rate compared with the ground truth.

In the clinical setting, a patient presenting with a suspected acute stroke in the ER will receive an initial assessment and CT scan. Before the radiologist drafts the report, it may be difficult and time-consuming for the ER physician and neurologist to read the CT images. As time is of the essence in critical events involving the brain, upon indication of IAT, the patient should receive IAT or be transferred to a medical center for IAT as soon as possible. Early diagnosis and prompt treatment are especially important for these patients. Ischemic stroke detection on NCCT is limited by its low sensitivity. The stroke-related hypoattenuation is subtle, the detection of which presents a distinct challenge to clinicians, especially in ordinary hospitals or hospital without an experienced stroke specialist. Figure 4 shows a stroke case with poor-visualized hypoattenuation in left caudate, putamen, internal capsule, insula, M1-2, and M4-6 in NCCT at early stroke change. Our DLAD model displays the infarcted area in ASPECT template. Furthermore, ASPECTS evaluation by visual inspection is a time-consuming process with interrater variability.

Previous studies have indicated the potential and clinical impact of auto-detection of ischemic stroke. One study reported on a classifier using a random forest model for automated ASPECTS scoring on NCCT images [6]. Meanwhile, Tang et al. developed a computer-aided detection of ischemic stroke using a novel circular adaptive ROI method to improve efficiency and accuracy in clinical practice [13]. The commercial software platforms providing some automated information are usually based on enhanced CT. The DLAD algorithm we propose herein could assist different subspecialists in ASPECTS evaluation on NCCT, thereby accelerating the assessment process and the time to transferal, or time to needle. Less than 2 min was the time required for one patient to use the DLAD algorithm to gain the automatic ASPECTS result, which may be faster than a radiologist to open the brain image in a long patient list. This DLAD algorithm may help physicians to accelerate the evaluation and treatment. It may further act as a warning process indicating that the radiologist takes precedence to read the AIS image. The DLAD algorithm may serve as a valuable tool to facilitate a prompt and efficient analysis based on NCCT.

## 5. Limitations

There are several limitations to the present study. First, this model did not receive potentially relevant clinical data including patient characteristics, stroke onset time, stroke severity, and treatment choice as parameters in the training process. These data points are important factors that may affect the prediction accuracy of the model. Second, the time interval between NCCT and DWI varied within one week. Thus, the restricted diffusion lesions on the DWI image may not have accurately represented the initial infarction lesion on the initial NCCT image. Third, the training dataset was limited. However, the adaptation of the proposed model to a multi-institute setting will form our future works and is beyond the scope of this paper. In the future, we will work to upgrade the DLAD algorithm by adding clinical structure data and will estimate a feedback platform in clinical practice. The NCCT images read by radiologists with annotated ASPECTS can be inputted into the dataset to retrain the DLAD algorithm.

## 6. Conclusions

The DLAD algorithm model proposed herein demonstrates a reasonable ability for ASPECTS scoring on NCCT images in patients presenting with AIS symptoms. The DLAD algorithm could improve clinicians’ performance and may serve as a facilitative tool to enhance and accelerate the decision-making process for front-line physicians.

## Figures and Tables

**Figure 1 jcm-11-05159-f001:**
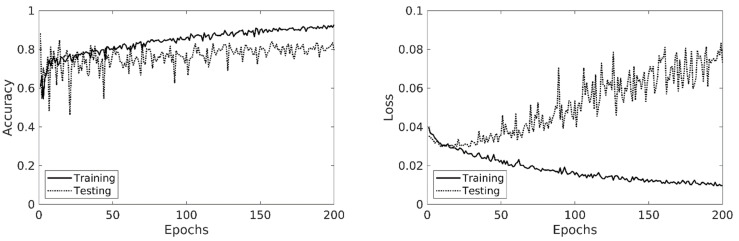
The accuracy and loss value over the training data after each epoch.

**Figure 2 jcm-11-05159-f002:**
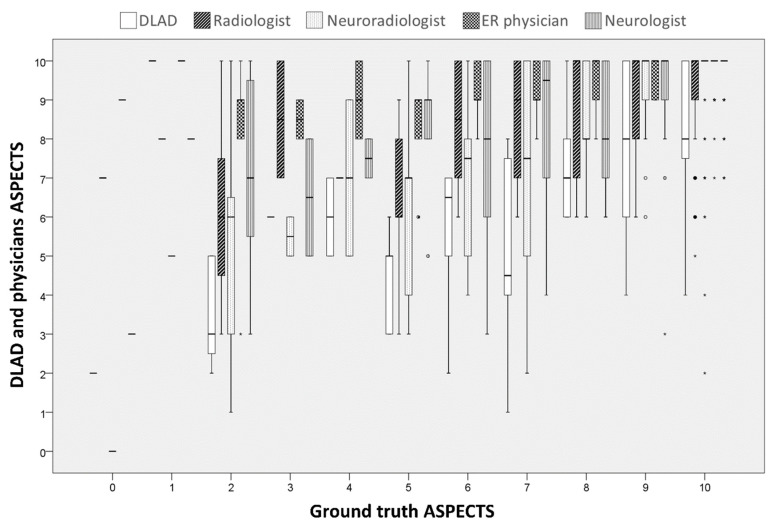
The boxplot shows the distribution of both human and the DLAD of ASPECTS scoring at each individual ground truth ASPECTS on NCCT. * Abbreviations: DLAD, deep learning–based automatic detection; ASPECTS, The Alberta stroke program early CT score; NCCT, non-contrast computed tomography.

**Figure 3 jcm-11-05159-f003:**
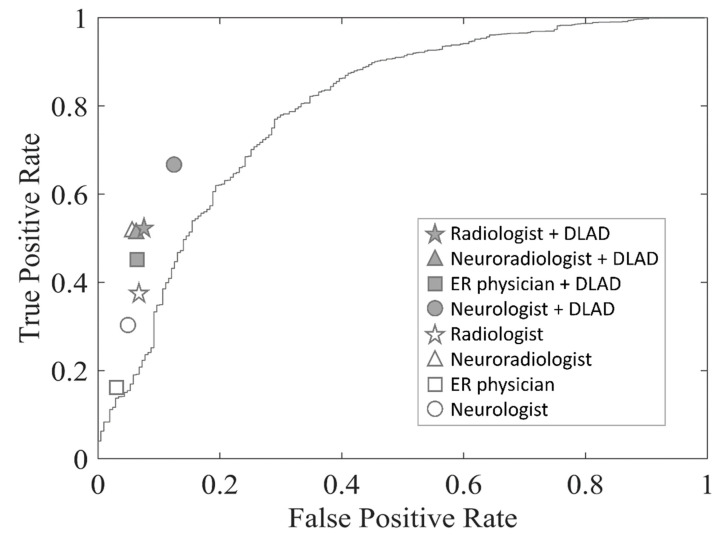
Comparison of diagnostic performance between DLAD algorithm and doctor groups. The ROC curves for DLAD in testing dataset. The performance of doctor with DLAD was significantly better with higher AUC values. Abbreviations: DLAD, deep learning–based automatic detection; ROC, receiver operating characteristic; AUC, area under the ROC curve.

**Figure 4 jcm-11-05159-f004:**
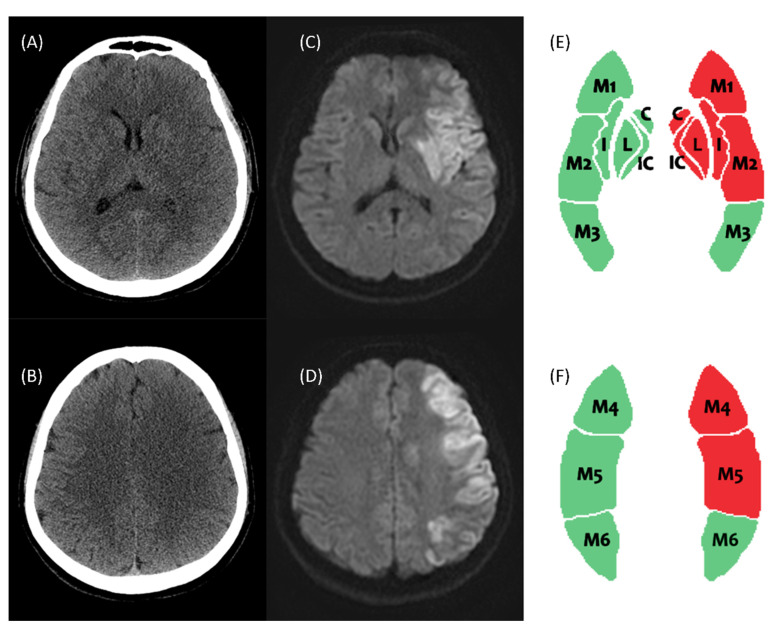
This is an early AIS case with poor-visualized signal change in NCCT. The NCCT shows faint hypoattenuation in left caudate, putamen, internal capsule, insula, and M1–2 at ganglionic level (**A**), and M4–6 at supra-ganglionic level (**B**). (**C**,**D**) The DWI image at the same levels shows restricted diffusion lesions. (**E**,**F**) Our DLAD model detects and displays the infarcted area in ASPECTS template. The red areas are the predicted infarction zone, whereas the green areas are the normal zone. Abbreviations: AIS, acute ischemic stroke; DWI, diffusion-weighted imaging; DLAD, deep learning–based automatic detection; ASPECTS, The Alberta Stroke Program Early CT Score; C, caudate; P, putamen; IC, internal capsule; I, insula.

**Table 1 jcm-11-05159-t001:** Detailed Architecture of the proposed model. s: stride. p: padding.

Modules	Components
Slice encoder	3D-Conv (8@3 × 3×1, s: 1 × 1 × 1, p: 1 × 1 × 0) ReLU
	3D-Conv (8@3 × 3 × 1, s: 2 × 2 × 1, p: 1 × 1 × 0) ReLU
	3D-Conv (16@3 × 3 × 1, s: 1 × 1 × 1, p: 1 × 1 × 0) ReLU
	3D-Conv (16@3 × 3 × 1, s: 2 × 2 × 1, p: 1 × 1 × 0) ReLU
	3D-Conv (32@3 × 3 × 1, s: 1 × 1 × 1, p: 1 × 1× 0) ReLU
	3D-Conv (32@3 × 3 × 1, s: 1 × 1 × 1, p: 1 × 1 × 0) ReLU
	3D-Conv (32@3 × 3 × 1, s: 2 × 2 × 1, p: 1 × 1 ×0) ReLU
Prediction aggregation	3DAdaptiveMaxPooling@4 × 4 × 20
	3D-Conv (16@1 × 1 × 20)
Classifier	3D-Conv (32@4 × 4 × 1, s: 1 × 1 × 1, p: 0 × 0 × 0), ReLU, Dropout (p:0.5)
	3D-Conv (1@1 × 1 × 1, s: 1 × 1 × 1, p: 0 × 0 × 0), Sigmoid

**Table 2 jcm-11-05159-t002:** The characteristics of training and testing datasets for the proposed model.

	Training Data (Mean ± SD)	Testing Data (Mean ± SD)
**Paient number**	168	90
**Age**	66.1 ± 11.8	70.1 ± 12.3
**Gender: male**	60%	69%
**NIHSS at ER**	15.94 ± 6.98	15.71 ± 6.67
**Pre-stroke mRS**	0.37 ± 0.94	0.43 ± 0.98
**Time to CT (min)**	259 ± 166	286 ± 209
**AIS (%)**	81.0%	62.2%

**Table 3 jcm-11-05159-t003:** The sensitivity, specificity, accuracy, precision, F1 score, kappa, and AUC of the DLAD for AIS on each ASPECTS region. (DLAD, deep learning–based automatic detection; AIS, acute ischemic stroke; ASPECTS, The Alberta stroke program early CT score).

Regions	Sensitivity	Specificity	Accuracy	Precision	F1 Score	Kappa	AUC
**Caudate**	40.9%	93.7%	87.2%	47.4%	0.439	0.367	0.770
**Putamen**	86.5%	55.9%	62.2%	33.7%	0.485	0.268	0.822
**Internal capsule**	64.3%	68.7%	68.3%	14.8%	0.240	0.130	0.691
**Insula**	87.1%	67.8%	71.1%	36.0%	0.509	0.351	0.878
**M1**	63.6%	95.3%	93.3%	46.7%	0.538	0.503	0.854
**M2**	76.2%	88.7%	87.2%	47.1%	0.582	0.511	0.868
**M3**	33.3%	85.2%	80.0%	20.0%	0.250	0.143	0.652
**M4**	72.7%	88.8%	87.8%	29.6%	0.421	0.366	0.832
**M5**	58.3%	80.1%	77.2%	31.1%	0.406	0.281	0.757
**M6**	38.9%	87.7%	82.8%	25.9%	0.311	0.217	0.638

**Table 4 jcm-11-05159-t004:** Performance of the DLAD for AIS as compared to physicians alone and physicians with DLAD. The sensitivity, specificity, accuracy, F1 score, kappa, and AUC on each ASPECTS interpretation at all ASPECT regions and for dichotomized ASPECTS (≥6 vs. <6) among the DLAD, doctor-read, doctor-read with DLAD, and ground truth ASPECTS. (DLAD, deep learning–based automatic detection; AIS, acute ischemic stroke; AUC, area under the ROC curve; ASPECTS, The Alberta stroke program early CT score).

All ASPECTS Regions	Sensitivity	Specificity	Accuracy	F1 Score	Kappa	AUC
DLAD algorithm	65.2%	81.6%	79.7%	0.43	0.32	0.73
Doctor-alone performance						
ER physician	15.9%	97.0%	87.7%	0.23	0.18	0.56
Neurologist	30.4%	95.4%	87.9%	0.37	0.30	0.63
Radiologist	37.2%	93.7%	87.2%	0.40	0.33	0.65
Neuroradiologist	51.7%	94.7%	89.7%	0.54	0.48	0.73
Doctor with DLAD performance						
ER physician	**44.9%** *	93.8%	88.2%	**0.47** *	**0.40** *	**0.69** *
Neurologist	**66.7%** *	87.8%	85.4%	**0.51** *	**0.43** *	**0.77** *
Radiologist	**52.2%** *	92.8%	88.1%	**0.50** *	**0.44** *	**0.72** *
Neuroradiologist	51.2%	94.1%	89.2%	0.52	0.46	0.73
**Dichotomized ASPECTS** **≥6 and <6**	Sensitivity	Specificity	Accuracy	F1 score	ICC	AUC
DLAD algorithm	72.2%	90.7%	88.9%	0.57	0.68	0.82
Doctor-alone performance						
ER physician	5.6%	100.0%	90.6%	0.11	0.19	0.53
Neurologist	27.8%	98.1%	91.1%	0.38	0.52	0.63
Radiologist	22.2%	99.4%	91.7%	0.35	0.50	0.61
Neuroradiologist	50.0%	95.7%	91.1%	0.53	0.65	0.73
Doctor with DLAD performance						
ER physician	**27.8% ***	96.3%	89.4%	**0.34** *	0.45	**0.62** *
Neurologist	**72.2% ***	90.1%	88.3%	0.55	0.67	**0.81** *
Radiologist	**50.0% ***	98.1%	93.3%	**0.60** *	**0.73** *	**0.74** *
Neuroradiologist	61.1%	95.1%	91.7%	0.59	0.71	0.78

* Permutation Test (10,000) with DLAD > without DLAD, one-tailed, *p* < 0.05.

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
