# Peer review of "Deep Learning-Based Automatic Detection of ASPECTS in Acute Ischemic Stroke: Improving Stroke Assessment on CT Scans"

_jcm, 2022, doi:10.3390/jcm11175159_

Round 1

Reviewer 1 Report

- to improve the quality of images, graphs and tables with captions and abbreviations. Moreover, Figure 2 could be integrated with a scatter diagram to better describe the results of both human and CNN diagnosis

- to separate Limitations and Conclusion paragraph

- to insert some brain images to demonstrate the burden methods to discern ictus

Reviewer 2 Report

The authors developed a deep learning-based automatic detection (DLAD) algorithm for ASPECTS in Acute Ischemic Stroke. I have several concerns:

-> There are no any recent studies cited in the manuscript. Is this because of the field is not getting attention anymore?

-> More sample size will increase the impact of the paper.

-> Deep learning model training should be explained in details. For example, any data augmentation techniques applied, architecture of the model, loss function, selected hyper parameters (batch size, learning rate).

-> DL models usually suffer from generalizability. One model trained on one institutional dataset performs poorly on images from another institution. Is this case for ur model? Is there any public datasets available to prove your model is generalizable? Having additional dataset from another source (public or other institutions) would greatly improve the significance of the paper.

-> The authors said that annotations were defined by experts. However, there could be inter reader variabilities among expert annotations. Do you think such variability affect the model performance?

-> Which method is employed for kappa score calculation?

-> Structure of the manuscript should be reorganized. I realized that some parts in the results section should be in the methodology actually.

Round 2

Reviewer 1 Report

Dear Authors,

you did a good job of reviewing. 

Reviewer 2 Report

The authors addressed all of my concerns. Thank you for this valuable study.